# Sustained enzymatic activity and flow in crowded protein droplets

Andrea Testa[1,5], Mirco Dindo[2,5], Aleksander A. Rebane[1], Babak Nasouri [3], Robert W. Style [1], Ramin Golestanian [3,4], Eric R. Dufresne[1✉] & Paola Laurino [2✉]

Living cells harvest energy from their environments to drive the chemical processes that enable life. We introduce a minimal system that operates at similar protein concentrations, metabolic densities, and length scales as living cells. This approach takes advantage of the tendency of phase-separated protein droplets to strongly partition enzymes, while presenting minimal barriers to transport of small molecules across their interface. By dispersing these microreactors in a reservoir of substrate-loaded buffer, we achieve steady states at metabolic densities that match those of the hungriest microorganisms. We further demonstrate the formation of steady pH gradients, capable of driving microscopic flows. Our approach enables the investigation of the function of diverse enzymes in environments that mimic cytoplasm, and provides a flexible platform for studying the collective behavior of matter driven far from equilibrium.

[1] Department of Materials, ETH Zürich, 8093 Zürich, Switzerland. [2] Protein Engineering and Evolution Unit, Okinawa Institute of Science and Technology Graduate University, 1919-1 Tancha, Onna 904-0495 Okinawa, Japan. [3] Max Planck Institute for Dynamics and Self-Organization (MPIDS), D-37077 Göttingen, Germany. [4] Rudolf Peierls Centre for Theoretical Physics, University of Oxford, Oxford OX1 3PU, United Kingdom. [5]These authors contributed equally: Andrea Testa, Mirco Dindo. ✉email: eric.dufresne@mat.ethz.ch; paola.laurino@oist.jp

The interior of cells can be highly crowded, with volume fractions ($\phi$) of about 20% for *E. coli*[1–5]. This means macromolecules cannot diffuse their own diameter without colliding with others. On top of these tight spatial constraints, a large fraction of these macromolecules are enzymes[6–9], which catalyze chemical reactions that release energy, creating transient mechanical stresses and chemical gradients. While this crowded and active milieu is an essential feature of the cytoplasm[10–14], we usually study the function of its molecular components, and even its collective behavior in dilute conditions, not very far from equilibrium. Working with dilute systems is an attractive alternative to working directly in the cytoplasm because it allows us to isolate the key elements that we want to study. On the other hand, without crowding and high metabolic densities, we fail to capture essential features of enzymes' physical and chemical niche.

In recent years, the molecular cell biology community has come to appreciate the essential role that membraneless organelles play in compartmentalizing the biochemistry of the cytoplasm[15–20]. These condensates of proteins and nucleic acids sequester enzymes and/or their substrates to regulate their activity[12]. Essential elements of membraneless organelles can be reconstituted in vitro, typically based on disordered proteins with weak–multivalent interactions[21]. Reconstituted and engineered condensates have been shown to be able to partition enzymes[14,22–28], and therefore their activity.

Here, we introduce a flexible approach to study enzymes in an environment that simulates the crowding and activity of the cytoplasm, while still being simple enough to understand and control. We exploit molecular crowding to generate dense liquid protein condensates that strongly partition enzymes, while allowing for unhindered diffusion of their small-molecule substrates and products. Loading droplets with model enzymes (L-lactate dehydrogenase (LDH) and urease) and dispersing them in a reservoir of the substrate, we achieve steady-state metabolic densities as high as any reported in cells. While the kinetics of urease are unaffected by compartmentalization, we observe a significant increase in the catalytic efficiency of LDH. For urease-loaded droplets, we observe a steady self-generated pH gradient, an essential feature of living systems. This pH gradient generates spontaneous flows within the droplets, reminiscent of cytoplasmic streaming.

Inspired by membraneless organelles within cells, we partition enzymes inside phase-separated protein droplets, as shown in Fig. 1a. A host protein is driven to form a membraneless droplet through crowding by a polymer[10]. This non-specific crowding not only drives the formation of droplets but will facilitate the efficient partitioning of enzymes into them[29]. Since small molecules are only weakly affected by the crowding agent, they can readily diffuse in and out of the droplet. In this way, trapped enzymes are easily fed by diffusion, and the product can rapidly diffuse out. In principle, an isolated droplet with a perfectly partitioned enzyme can maintain arbitrarily high metabolic rates without running out of the substrate. To avoid significant local heating and minimize gradients of activity across droplets, they should be on the micrometer scale, see discussion in the Supplementary Materials and refs. [30,31].

## Results and discussion

**Droplet system.** We demonstrate this general approach using bovine serum albumin (BSA) as the host protein and 4 kDa polyethylene glycol (PEG) as a crowding agent. In our standard conditions, we prepare a solution with average concentrations of 232 mg/mL PEG and 37 mg/mL BSA in potassium phosphate buffer, and the system spontaneously separates into two phases, a BSA-rich droplet phase ($434 \pm 7$ mg/mL BSA and $25 \pm 3$ mg/mL

PEG) and a PEG-rich continuous phase ($243 \pm 4$ mg/mL PEG and $13 \pm 1$ mg/mL BSA). In these conditions, the volume fraction of the BSA-rich phase was determined to be $\phi = 3 \pm 2$%, as described in Supplementary Materials. The full phase diagram of this BSA-PEG system is shown in Fig. 1b. With this system, we can vary the concentration of BSA in the droplets from about 350 to 500 mg/mL. Note that these are somewhat higher than the values reported for the concentration of protein in typical cytosol[32–35]. The viscosity of the droplet phase was determined by particle tracking microrheology to be 2.1 Pa s, about 2000× that of water (Fig. S1), and comparable to values reported for the cytoplasm[36,37]. Thus, proteins in the droplet phase are highly crowded in a manner similar to the cytoplasm. Note that the two phases can be separated by centrifugation, which allows us to adjust $\phi$ at will by diluting the BSA-rich phase with the desired quantity of the PEG-rich phase.

**LDH partitioning and catalytic efficiency.** As a proof of concept, we chose to work with the enzyme L-lactate dehydrogenase (LDH), whose substrate and product are pyruvate and lactate, respectively[38] (Fig. S2). Fluorescent imaging of tagged LDH (Fig. 1a) qualitatively shows that it partitions well to the droplet phase. To quantify this with an unlabeled enzyme, we prepare the droplet and continuous phases equilibrated in the presence of an enzyme. Then, we compare lactate production at $\phi = 3$% and $\phi = 0$% (where droplets have been removed by centrifugation). The global rate of lactate production at $\phi = 3$% is about 100× faster than $\phi = 0$%, estimated from the early time lactate production rate in Fig. 1c. Thus, LDH activity is strongly partitioned to the droplet phase.

With these results, we can now compare the kinetics of LDH in the BSA-rich phase and in buffer (Fig. 1d). At all substrate concentrations, the LDH reaction velocity, $V$, is higher in the droplets than in the buffer. Michaelis–Menten parameters are reported in the inset of Fig. 1d (obtained from fitting data in Fig. S3 using pyruvate concentrations from 0 to 1.3 mM). While the $K_m$ values are nearly identical, $k_{cat}$ increases significantly. Expressed in terms of catalytic efficiency, we find $k_{cat}/K_m = 2131 \pm 520$ s$^{-1}$ mM$^{-1}$ in the two-phase system and $k_{cat}/K_m = 1229 \pm 276$ s$^{-1}$ mM$^{-1}$ in buffer, the latter in agreement with literature values[39]. Note that the catalytic efficiency in the supernatant is identical to plain buffer (Fig. S3). Compartmentalization and crowding, therefore, lead to a significant enhancement of the kinetics of LDH. Note that the reaction velocities decrease above 1.3 mM pyruvate concentration (Fig. 1d). This substrate inhibition effect is well documented for LDH in buffer[40,41] and is characterized by the inhibition constant, $K_i$, which we find to be unchanged by compartmentalization.

The rate of consumption of chemical energy inside the droplets, $\dot{Q}$, determined by the standard enthalpy of the reaction and the measured reaction rates, is shown in Fig. 1e. With metabolic densities approaching 1 MW/m$^3$ (Figs. 1e and S4), these droplets exceed the metabolic rates of even the most voracious unicellular organisms[42].

Reducing the volume fraction of the droplets 3000× to $\phi \approx 10^{-5}$, the reaction can run steadily for more than 1 h (Fig. 1f, g). By contrast, a simple LDH solution at an enzyme concentration equal to that present in the droplets would consume all the pyruvate in less than 5 s (Fig. 1f, inset). Thus, partitioning enables a thousand-fold increase in a lifetime for experiments with concentrated enzymes.

**Imaging urease activity.** To directly visualize the localization of the enzymatic reaction to the droplet, we switched from LDH to urease. This enzyme hydrolyzes urea to produce carbon dioxide

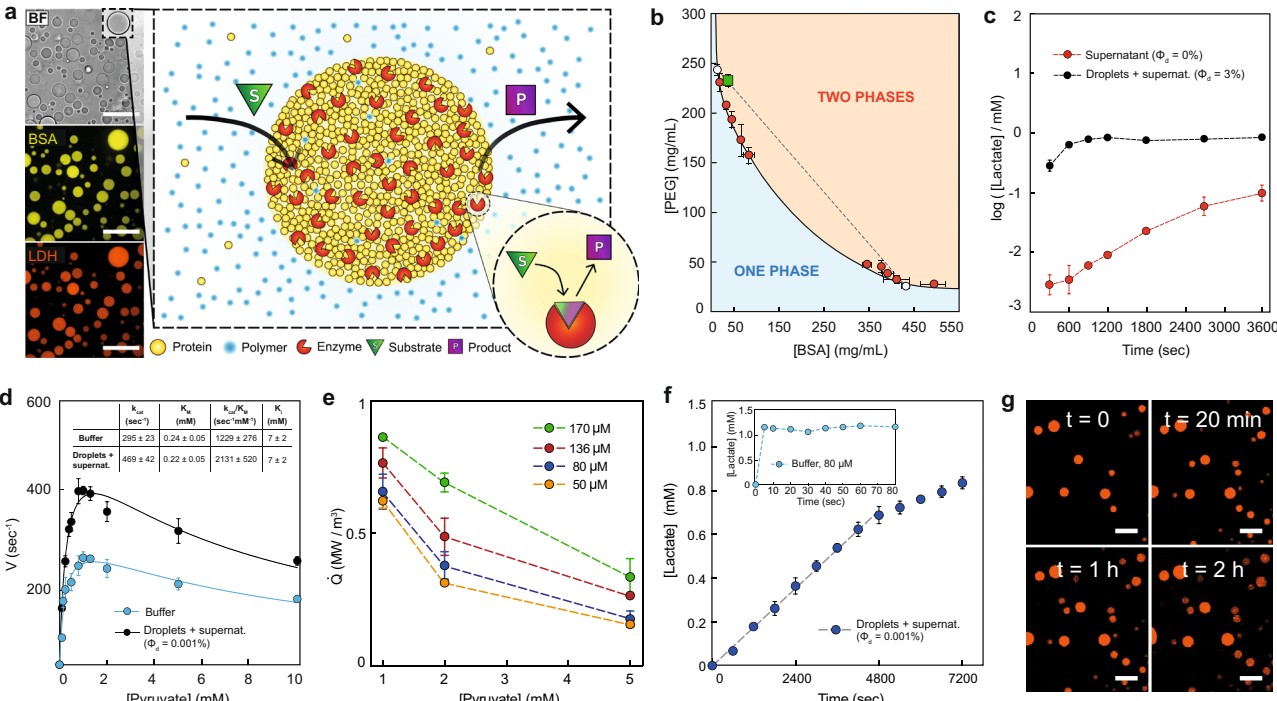

**Fig. 1 Liquid-liquid phase separation of BSA droplets and LDH metabolic activity. a** Schematic representation of a general active liquid-liquid phase-separated protein droplet system with partitioned enzyme, by the action of a crowding polymer. The substrate is present in the continuous phase and can freely diffuse inside the droplets, while the product formation catalyzed by the enzyme in the droplet phase can diffuse out. The relative concentrations are not drawn to scale. Left side: confocal microscopy images of droplets with labeled protein (BSA) and enzyme (LDH). Top to bottom: bright-field channel, fluorescent BSA, and LDH channels. Confocal images were acquired for more than 20 independent experiments with similar results. **b** Phase diagram of the PEG-BSA phase-separated droplets. The green square denotes the overall composition of the droplet suspension at the chosen working condition. White circles represent the compositions of the two phases at the working condition. The dashed line indicates tie line connecting these two compositions. Data are represented as mean values ± standard deviation (SD) from three independent experiments ($n = 3$). **c** Lactate production in a $\phi = 3\%$ dispersion of droplets in the supernatant with an LDH concentration of 0.060 μM in the droplets in the presence of 1 mM pyruvate and 2 mM NADH (black) and in the same system after the droplet phase was removed by centrifugation (red). See SI for the calculation of enzyme concentration inside the droplets. Data are represented as mean values ± SD from three independent experiments ($n = 3$). **d** Velocity values at different pyruvate concentrations of droplets containing 3.3 μM LDH, dispersed in the supernatant (black) compared to that obtained in buffer using 2 nM LDH (cyan). The velocity values have been fitted using the substrate inhibition equation[85]. Inset of panel **d**, kinetic parameters of droplets containing-LDH compared to those of free LDH in a buffer. The values of $k_{cat}$ and $K_M$ have been calculated by plotting the velocity values from 0.1 to 1.3 mM pyruvate (values without substrate inhibition) using the Michaelis–Menten equation[86]. The graph is reported in Fig. S3. Data are represented as mean values ± SD from three independent experiments ($n = 3$). **e** Different metabolic rates obtained varying the enzyme (concentration of LDH in the droplets 50, 80, 136, and 170 μM respectively) and substrate concentrations (1, 2, and 5 mM pyruvate) in the presence of 10 mM NADH. The volume fractions ($\phi_D$) used in these experiments and the trends of lactate production over time at different enzyme and pyruvate concentrations are reported in Fig. S4. Data are represented as mean values ± SD from three independent experiments ($n = 3$). **f** Lactate production over time (equal to pyruvate consumption) for LDH in droplets (concentration 80 μM inside the droplets) in presence of 5 mM NADH and 1 mM pyruvate. Inset: Lactate production over time of LDH in buffer, at a concentration equal to that in droplets (80 μM), in the presence of 2 mM NADH and 1 mM pyruvate. **g** Confocal images of the droplets partitioning labeled LDH during a sustained activity at different times, at the same conditions (LDH, pyruvate, NADH concentrations, and volume fraction) as in panel **f**. Confocal images were acquired for more than ten independent experiments with similar results. All scale bars are 50 μm.

and ammonia, a strong base[43] (Fig. S2). Thus, the local effect of urease can be visualized using a pH-sensitive fluorescent dye. Before imaging the localization of the reaction, we checked the partitioning of urease to the droplets (Fig. 2a) and characterized its reaction kinetics macroscopically. For unlabeled urease, macroscopic measurements revealed that the rate of ammonia production was more than 400-fold higher at $\phi = 0.03\%$ than $\phi = 0\%$, as shown in Fig. 2b. This implies that the reaction, as it was for LDH, is strongly partitioned to the droplet phase. However, we notice that the catalytic efficiency of urease is slightly decreased in our droplets compared to our standard buffer conditions (Fig. 2c). Specifically, $k_{cat}/K_m = 781 \pm 107$ s$^{-1}$ mM$^{-1}$ in the droplets and $2000 \pm 290$ s$^{-1}$ mM$^{-1}$ in standard buffer[44]. Notably, $K_m$ increases threefold in the droplets phase (Fig. 2c) while the $k_{cat}$ slightly increases (Table inset in Fig. 2c). However,

$k_{cat}$ and $K_m$ in the supernatant phase are identical to their value within the droplets (Fig. S5). Therefore, confinement and crowding has no significant effect on urease kinetics.

To directly visualize local changes in pH we used a pH-sensitive dye (SNARF-1) (Fig. S6). In the absence of urea (Fig. 2d, control), the pH inside the droplets over time is stable at the buffered value of 7.0–7.2. On the contrary, adding 50 and 100 mM urea, the local pH inside the droplets increases over time to a plateau around pH 8.5–9.0, as shown in Fig. 2d and in the same timescales, the global pH of the reaction mixture increases only slightly from 7.2 to 7.3 (Fig. S7). These experiments confirmed that the local pH inside the droplets is changing quickly while the global pH of the reaction mixture is stable. Thus, compartmentalization of urease into droplets creates a stable pH gradient. Interestingly, the pH change inside the

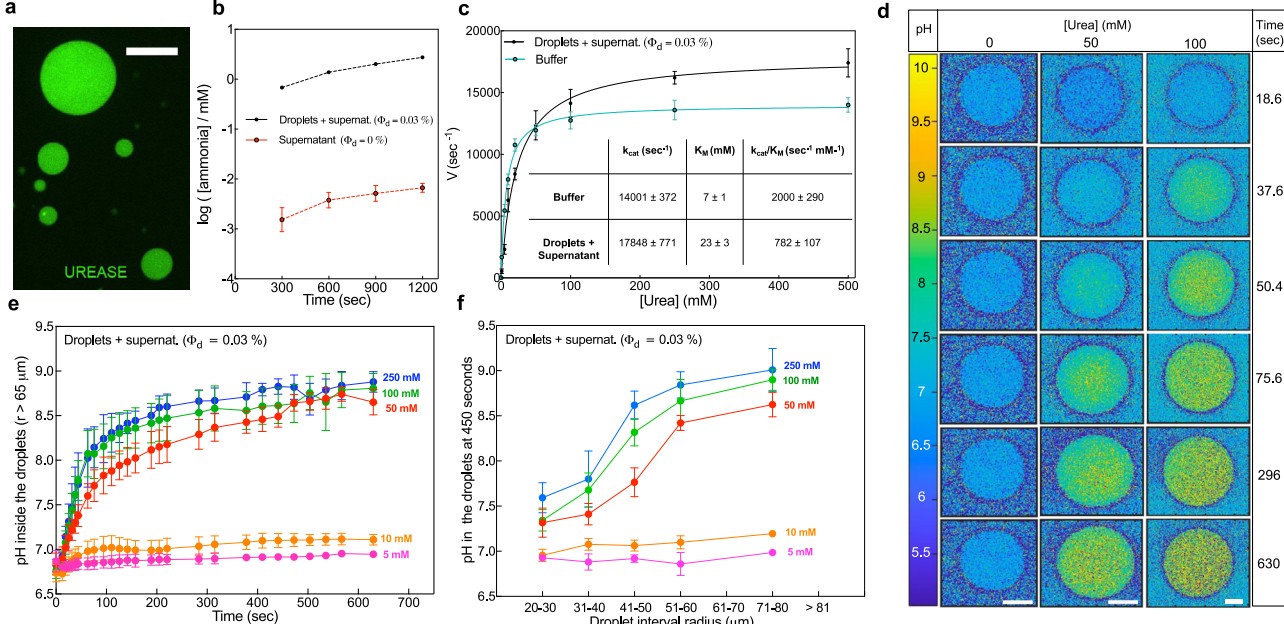

**Fig. 2 Evaluation of partitioning, kinetic characterization, and pH change generated by urease activity inside the droplets. a** Confocal microscope image of Alexa Fluor 488-labeled urease at 1.0 μM inside the PEG-BSA droplets (scale bar is 20 μm). **b**, **c** Urease activity measurement in the droplets and supernatant and in the supernatant only and Michelis–Menten curves of urease in the droplets and supernatant and in the buffer. Inset of panel **c**, kinetic parameters values of droplets containing-urease compared to those of free urease in the buffer. The concentration of urease in the activity measurements is 1 μM in the droplets and supernatant and 0.3 nM in a buffer. Data are represented as mean values ± SD from three independent experiments ($n = 3$). **d** Visualization of the pH change in the droplets containing 1 μM urease at different times using two different concentrations of substrate (50 and 100 mM urea) along with the control (no substrate). Scale bars are 50 μm. **e** Evaluation of the pH change over time for droplets with a radius above 65 μm containing 1 μM urease at different urea concentrations. Data are represented as mean values ± SD from three independent droplets ($n = 3$). **f** pH change inside the droplets at 450 s at different urea concentrations and 1 μM urease. Data are represented as mean values ± SD from five isolated droplets ($n = 5$).

droplets is not only dependent on the urea concentration (Fig. 2e) but also the droplet radius (Fig. 2f and Fig. S8). The plateau pH inside the droplets increases with droplet radius, suggesting significant transport limitations (Fig. 2f).

**Flow in urease droplets**. To characterize small molecule transport in the urease-loaded BSA droplets, we added rhodamine-B to the continuous phase. In the presence of urea, transport of the dye into the droplet was asymmetric, suggesting advection, Fig. 3a. Time-lapse imaging of partitioned fluorescent nano-particles revealed an underlying flow, with a magnitude of about 0.1 μm/s (Fig. S9). This internal flow is modified by the presence of nearby droplets shown by the time-lapse images in Fig. 3b and in the Supplementary Movie 1. Flow across the center of each droplet points toward its neighbor. This suggests that concentration gradients created by enzymatic activity could drive flow in nearby droplets. Since we observe flow only with active urease-loaded droplets, we hypothesized that this flow could be due to local pH gradients.

To test this, we generated a pH gradient by releasing supernatant adjusted to pH 8.4 and tagged with fluorescein isothiocyanate (FITC) from a micropipette close to the droplets. We observed similar flow patterns to those exhibited by the active droplets, directed toward the pipette (Fig. 3c and Fig. S10).

To quantify the coupled flow of adjacent droplets, we tracked the particle trajectories shown in Fig. 3b. Exploiting the steady nature of the flow, we calculated the velocity within the droplet at each point on a grid. The resulting velocity field, shown as gray arrows in Fig. 3d, indicates fluid speeds from zero to 0.15 μm/s, comparable to velocities observed during cytoplasmic streaming, e.g., ref. [45]. Interestingly, the sharpest gradients in the fluid

velocity are found near the edges of the droplets (Figs. S11 and S12). Combining this information with the droplet viscosity, we determined the apparent shear stresses at the droplet interface, which have a magnitude of 10 mPa (red arrows in Fig. 3d).

We identify two distinct mechanisms that could underlie the observed flow: a classical mechanism involving the surface, and a novel mechanism driven in the bulk. Both provide qualitatively similar flow profiles within the droplet (see Supplementary Materials for details). In the first mechanism, gradients of pH create gradients of the interfacial tension between the droplet and continuous phases, $\gamma$[46]. This Marangoni effect gives a characteristic velocity $(\partial\gamma/\partial c)\Delta c/\eta_{in}$. Here, $c$ is the concentration of the pH-determining species and $\Delta c$ is the scale of its difference between the source and buffer. In the second mechanism, proteins in the bulk create a diffusiophoretic flow throughout the droplet, due to the confining effects of the densely-packed proteins and the surrounding medium, reminiscent of the osmotic flows due to noncontact interactions[47]. In this mechanism, the velocity scale is given by $\xi k_B T R \Delta c/\eta_{in}$, where $\xi$ captures the effective confining force experienced by each protein. Related diffusiophoretic effects have recently been shown to lead to protein organization and transport through the establishment of chemical gradients, via diffusiophoresis[48,49]. Interestingly, the two velocity scales depend differently on the droplet radius, favoring the bulk-driving mechanism for larger droplets. A third mechanism driven by slip at the surface of the droplet could also contribute to the flow inside the droplet. However, we expect that this effect is orders of magnitude smaller than the other two (see Supplementary Materials for details).

To test the feasibility of pH-driven Marangoni flows, we inferred surface tension differences along the droplet interface

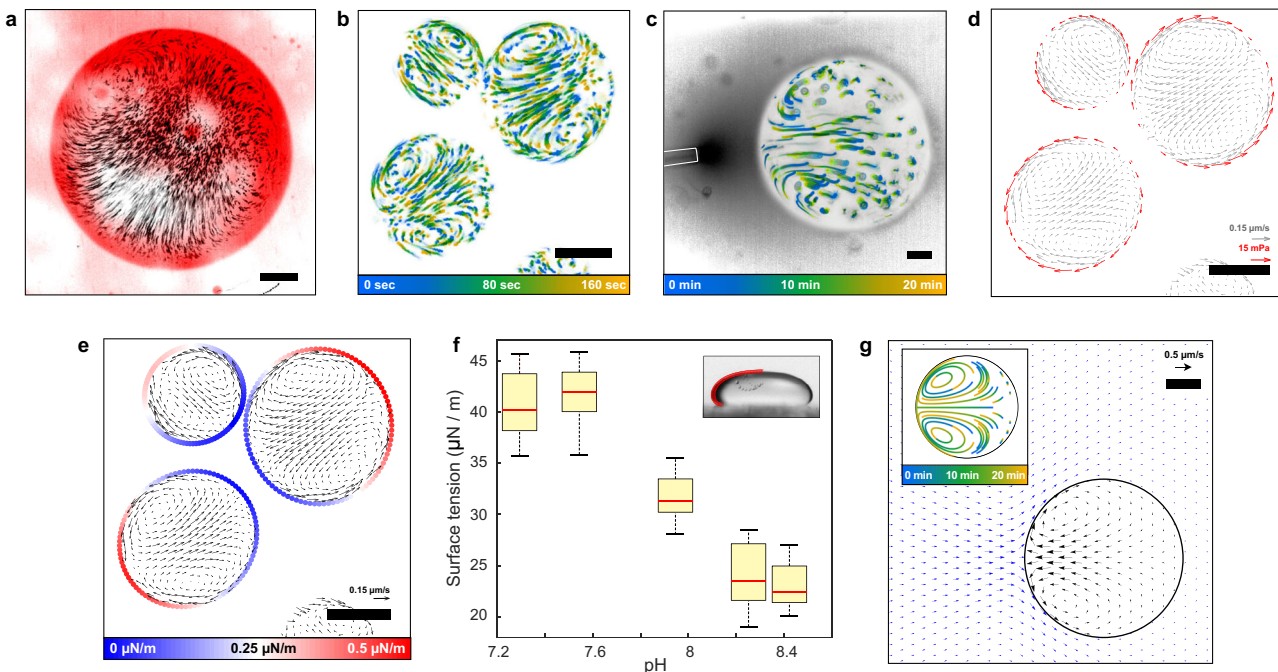

**Fig. 3 Activity-induced flow. a** Fluorescent intensity of rhodamine diffusing in an active droplet (1 μM urease in the droplet, 100 mM urea) after 200 s from addition (red), overlapped on the time projections of fluorescent tracers (black). **b** Time projection of fluorescent intensity for a group of three active droplets (1 μM urease in the droplet, 100 mM urea). The tracks show the color-coded trajectories of fluorescent tracers. **c** Time projection of fluorescent intensity for a droplet and micropipette. The tracks show the color-coded trajectories of fluorescent tracers, while the dark halo the released solution. **d** Shear stress vectors calculated at the droplets' edges (red arrows) overlapped on the fluid velocity field (gray arrows) for the droplets in panel **b**. **e** Delta surface tension integrated along the droplets' edges overlapped on the fluid velocity field (black arrows) for the droplets in panel **b**. **f** Box plot of surface tension difference between the supernatant and droplet phase as a function of the pH of the supernatant, calculated with the sessile drop method[50] (inset). The central mark indicates the median, the top and bottom edges of the box the 25th and 75th percentile, the whiskers the maximum and minimum value of the data. For each point, 20 individual droplets were analyzed ($n = 20$). The droplet in the inset has a radius, at its widest point, of 600 μm. **g** The induced flow field inside (black arrows) and outside (blue arrows) of the droplet in the micropipette experiment, obtained from the theoretical model (see SI for details). Time projection of the flow trajectory inside the droplet is shown in the inset. All the scale bars are 20 μm.

and quantified the pH dependence of the surface tension. Integration of the shear stresses along the droplets' edge gives us the relative surface tension at each point along the interface (Figs. 3e and S13). The inferred surface tension is lower close to neighboring droplets, and the magnitude of the surface tension differences across the droplet is about 0.5 μN/m. To determine whether these surface tension differences could be driven by pH, we applied the sessile drop method[50] to measure the equilibrium surface tension between the droplet and continuous phases at different values of pH (Fig. S14). We observed a significant reduction of the surface tension from roughly 40 μN/m at below pH 7.6 to 23 μN/m above pH 8.2 (Fig. 3f). These alkaline pHs are readily achieved during the urease reaction (Fig. 2), and fit with the observation of reduced surface tensions for nearby droplets.

A quantitative comparison of theory and experiment requires detailed information on the pH profile, including the continuous phase. For simplicity, we focused on the micropipette experiment and ignored the action of the buffer in stabilizing the pH value. In this approximation, the concentration of pH-determining species falls off roughly like $1/r$, and the predicted velocities from a pure surface tension-driven flow are 100-fold too large, as discussed in the Supplementary Materials. This, however, could be countered by diffusiophoretic flow to generate velocity scales comparable to those observed in the experiments, as shown by the simulation of the micropipette experiment in Fig. 3g.

We have shown that enzymatic reactions can be strongly compartmentalized into crowded protein-rich droplets, reaching steady metabolic rates that are as high as any reported in a living system. Interestingly, LDH assays show a significant increase in catalytic efficiency in the droplets suggesting that crowding and confinement might have unappreciated effects on enzymatic activity. Furthermore, we create steady pH gradients, mimicking an essential feature of prebiotic conditions[51–53]. Generally, the free energy stored in these pH gradients are capable of doing work. Specifically, we showed enzyme-generated pH gradients can drive steady flow within droplets, mimicking cytoplasmic streaming[54,55].

Our work opens a number of new research directions. In biochemistry, the effects of crowding and high metabolic density on enzyme activity and protein folding are a paramount challenge[56–59]. Our approach facilitates metabolic engineering, through the compartmentalization of enzymatic cascade reactions[60–63]. Furthermore, the biocatalysis applications compartmentalization are vast, including continuous biochemical synthesis of small molecules in droplet microreactors[29,64].

Further, our approach provides a flexible platform for studying materials driven far from thermodynamic equilibrium, yet in steady-state. At the continuum scale, it enables investigations of how activity can affect material properties, and drive new types of flow. At the molecular scale, the ability to create strong concentration gradients in a steady-state will enable a mechanistic understanding of the emerging phenomenon of enzyme chemotaxis[65–67]. The enzymatic activity might drive novel behaviors, including the emergence of early metabolic pathways, motility, or division[68,69].

Our active droplets may also serve as microscopic "test tubes" for the reconstitute of higher-order biological function, including the studies on the origin of life[70–75]. Experimental studies of primitive

protocell models based on compartmentalization[76–83] have advanced tremendously in recent years and complement the present work.

## Methods

**Materials**. Polyethylene glycol (PEG) 4000 Da (A16151) was purchased from Alpha-Aesar. Bovine serum albumin (BSA) (A7638), Potassium phospate dibasic trihydrate (60349), 3-(Trimethoxysilyl)propylmethacrylate (440159), 2-Hydroxy-4′-(2-hydroxyyethoxy)-2-methylpropiophenone (410896), Poly(ethylene glycol) diacrylate (PEGDA) 700 Da (455008), L-lactate dehydrogenase (LDH) from muscle rabbit (L1254), Jack bean urease (U4002), β-Nicotinamide adenine dinucleotide, reduced disodium salt hydrate (NADH) (N8129), Pyruvate (P8524), Lactate assay Kit (MAK064), Phenol nitroprusside solution (P6994), Alkaline hypochlorite solution (A1727), Urease Activity Assay Kit (MAK120), Rhodamine-B (R6626) were purchased from Sigma-Aldrich. Potassium phosphate monobasic (42420) was purchased from Acros Organics. Deuterium oxide (D₂O) (DE50B) was purchased from Apollo. N,N-dimethylformamide (DMF) (D119) was purchased from Fisher Chemicals. Alexa Fluor 594 and 488 Microscale Protein Labeling Kit (A30008 and A30006) and Carboxylic acid, Acetate, Succinimidyl Ester SNARF-1 (S2280) were purchased from Thermo Fischer Scientific. Trichloroacetic acid (TCA) (34603) was purchased from Nacalai Tesque. Fluorescent polystyrene nanoparticles tracers of 0.2 and 1 μm of diameter (FCDG003, FCDG006) were purchased from Bangs Laboratories. Fluorescein isothiocyanate (FITC) (AB178737) was purchased from abcr.

**Droplet formation**. A polyethylene glycol stock solution at 600 mg/mL was prepared mixing the appropriate amount of PEG 4000 Da and Milli-Q water. The pH of the solution was measured to be 7 with a pH meter (Orion Star A111, Thermo Fisher and F71, HORIBA Scientific).

A bovine serum albumin stock solution with a target concentration of 5 mM (332 mg/mL) was prepared mixing the appropriate amount of BSA and Milli-Q water. The actual concentration was confirmed by measuring the absorption intensity at the 280 nm peak with a UV-vis spectrophotometer (Cary 60 Spectrophotometer, Agilent Technologies, and UV-1900 UV-Vis Spectrophotometer, Shimadzu), assuming $\varepsilon_{280} = 43,824 \, M^{-1} \, cm^{-1}$ (https://web.expasy.org/protparam/). The pH of the solution was measured to be 7.

A potassium phosphate buffer (KP) stock solution at 500 mM and pH 7 was prepared.

The droplet suspensions were prepared mixing appropriate amounts of PEG, BSA stocks, together with KP buffer, KCl, and Milli-Q water in an Eppendorf tube. At the standard working conditions, we mixed the components with a final target concentration of 230 mg/mL of PEG, 30 mg/mL BSA, 200 mM KCl, and 100 mM KP pH 7.0. Additional components like enzymes, cofactors, substrates, and fluorescent nanoparticles were also added to the tube when needed. The tubes were gently shaken by hand to promote the mixing. When required, the supernatant was isolated centrifuging the droplet suspension for 30 min at 16,900x*g* at room temperature and extracting the top phase with a pipette.

**Phase diagram**. The phase diagram was mapped as a function of BSA and PEG composition. Specifically, we determined the two arms of the binodal by measuring the concentrations of BSA and PEG in both droplet and supernatant phases for various average sample compositions. All samples were prepared and measured in triplicate at room temperature. The two phases were obtained by centrifuging the phase-separated samples for 60 min at $16,900 \times g$ and then carefully transferring the supernatant to a separate tube using a micropipette. The centrifugation step was repeated and any residual supernatant was removed from the droplet phase.

The BSA concentrations in the two phases were determined by measuring absorbance at 280 nm as explained in the previous Methods section (Droplet Formation). Measurements were performed in a quartz cuvette (UQ-124, Portmann Technologies) on samples diluted with Milli-Q water (typically 200-fold diluted for droplet phase and 10- to 40-fold diluted for supernatant phase).

PEG concentrations were determined using proton nuclear magnetic resonance (¹H NMR) spectroscopy[84] with D₂O as solvent. DMF was added to the sample at an appropriate final concentration (10–660 mM) to serve as a calibration standard for the PEG concentration measurement. Measurements were performed in disposable 5 mm NMR tubes (Type E, Schott) on samples diluted with D₂O (20-fold diluted for droplet phase and 200-fold diluted for supernatant phase). ¹H NMR spectra were acquired using a 300 MHz instrument (300 Ultrashield, Bruker) at 298 K and averaged 16 times. The resulting data were analyzed using the MestreNova software (Mestrelab Research). The peaks at 3.02 ppm and 2.86 ppm stem from the two methyl groups in DMF; their integrals represent the signal from six protons per DMF molecule. The PEG peak appears at 3.71 ppm and its integral represents the contribution from ~354 protons per PEG molecule. The PEG concentrations were thereby determined from the relative height of the 3.71 ppm peak to the two DMF peaks at 3.02 ppm and 2.86 ppm and the known DMF concentration in the sample. Raw NMR data is included as Supplementary Data.

**Volume fraction**. The volume fraction of the droplet phase at the chosen working conditions was determined using two independent methods. In the first method, we applied the lever rule to the previously measured BSA and PEG compositions of the droplet and supernatant phases, as well as the average composition of the droplet suspension. Specifically, the lever rule gives the droplet volume fraction, $\phi$, as

$$\phi = \frac{c_{a,i} - c_{S,i}}{c_{D,i} - c_{S,i}}, \tag{1}$$

where $c_{D,i}$ and $c_{S,i}$ are the mass concentrations of component $i$ in the droplet and the supernatant phases, respectively, $c_{a,i}$ is the average concentration of component $i$ in the droplet suspension, and $i \in \{BSA, PEG\}$. Using this formula, the droplet volume fraction can be calculated either from BSA or PEG concentration measurements. Because of the vicinity to the binodal line, we found that small variations of the average BSA concentration had a strong influence on the droplets' volume fraction, yielding a range from ($1\% < \phi_d < 5\%$). For our calculations, we used the average value $\phi = 3\%$.

In the second method, we prepared several mL of the droplet suspension, and separated the two phases by centrifugation and directly compared the volumes of the two phases. We found $\phi$ values consistent with the lever rule.

**Protein labeling**. L-lactate dehydrogenase and Jack bean urease were tagged using Alexa Fluor 594 and 488 Microscale Protein Labeling Kit respectively, using the manufacturer's protocol but repeating the dialysis step three times to remove the free dye in the protein solution. The BSA in Fig. 1a was tagged with Alexa Fluor 647 Microscale Protein Labeling Kit using the same purification procedure. The reaction between SNARF-1 and BSA was carried out in potassium phosphate 0.1 M pH 7.0 in presence of sodium bicarbonate 0.1 M for at least 1 h at room temperature. Tagged proteins were further purified using membrane dialysis to reduce the free probe in solution. To determine the protein concentration of LDH and urease we used $\varepsilon_{280nm} = 43,680 \, M^{-1} \, cm^{-1}$ and $\varepsilon_{280nm} = 54,165 \, M^{-1} cm^{-1}$, respectively (https://web.expasy.org/protparam/).

**Statistics and reproducibility**. No statistical method was used to predetermine sample size. No data were excluded from the analyses. The experiments were not randomized. The Investigators were not blinded to allocation during experiments and outcome assessment.

**Reporting Summary**. Further information on research design is available in the Nature Research Reporting Summary linked to this article.

## Data availability
The datasets generated during and/or analysed during the current study are available from the corresponding author on reasonable request.

## Code availability
Code used in the current study are available from the corresponding author on reasonable request.

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

## Acknowledgements

This work was supported by grant number 172824 from the Swiss National Science Foundation to ERD, grant number GR19106 from the Japan Society for the Promotion of Science (JSPS) as part of the Bilateral Swiss-Japanese Science and Technology Program to AT. Financial support by the Okinawa Institute of Science and Technology to P.L. is gratefully acknowledged. M.D. thanks the Japan Society for the Promotion of Science (JSPS). Fellowship number: P19764. R.G. acknowledges support from the Max Planck School Matter to Life and the MaxSynBio Consortium which are jointly funded by the Federal Ministry of Education and Research (BMBF) of Germany and the Max Planck Society. We thank Paolo Barzaghi from the imaging Section of Okinawa Institute of Science and Technology (OIST) for the help with confocal microscopes and Kieran Deasy from the engineering section of OIST for 3D printing the well chambers. We thank Mahesh Bandi, Dan S. Tawfik, Anna Magdalena Klarkowska, Yingjie Xiang, Andreas Küffner, Hendrik Spanke, Dominic Gerber, Alessandro Bevilacqua, and Riccardo Martini for useful discussions.

## Author contributions

Sustained activity concept and design: E.R.D. and A.T.; Protein crowding concept and design, enzyme activity concept, enzyme selection, enzyme kinetics, and data analysis: M.D. and P.L.; Protein labeling: M.D. and A.A.R.; Enzymes activity assays set up, enzyme partitioning assay: M.D.; BSA-PEG system design and development: A.T., M.D., P.L., and E.R.D.; Sample chamber design, micropipette experiment: A.T.; Substrate addition protocol: A.T. and M.D.; Phase diagram: A.A.R., M.D., E.R.D., and P.L.; Rheology and microrheology: A.T.; Heat, substrate, and product transport model: E.R.D., A.T., and A.A.R.; pH imaging: M.D.; pH image analysis: M.D., A.T., and E.R.D.; Particle tracking velocimetry: A.T., E.R.D., and R.W.S.; Advection measurements: A.T. and M.D.; Flow field analysis: R.S., A.T., and E.R.D.; Fluid flow theory concept and design: R.G. and B.N.; Surface tension measurements: A.T. and A.A.R.; Paper writing: E.R.D., P.L, R.G, A.T., and M.D. with input from all.

## Competing interests

The authors declare no competing interests.
