## [Peer Review File · Nature Communications]

Reviewers' Comments:

Reviewer #1:

Remarks to the Author:

Review for Sustained Enzymatic Activity and Flow in Crowded Protein Droplets

The manuscript Sustained Enzymatic Activity and Flow in Crowded Protein Droplets from Testa and colleagues is an exciting examination of the effects of crowding on enzymatic activity. Specifically, they show that when proteins are crowded with high levels of macromolecules so that they phase separate into liquid droplets, enzymes can partition into the droplets to have sustained activity. While the overall concentration of the enzyme in the test tube is low, the concentration within the droplets is higher. Further, the activity is sustained because of the partitioning of the enzymes and the diffusion-limits for substrate to get to the proteins. Even more exciting, they show that these partitioned enzymes can produce work and flows. These flows are driven by gradients created across the boundary of the droplets. Thus, the droplets confine proteins and small molecules without the need for a membrane. As the author's point out, these condensates could be the basis of early life and can also serve as platforms for chemical and materials innovation.

Positive aspects of the manuscript were:

1. Very exciting science and concepts
2. Clear experimental results that were highly quantitative
3. Modeling to accompany the experiments to help elucidate possible mechanisms of the flow induced in the droplets

Some issues that I found with the manuscript were:

1. Some aspects of the manuscript were not clearly described or were described at too high a level for the broad audience of the journal
2. Lack of information to assess reproducibility - repeats, error bars, fits, etc.

Below, I outline some of the issues I found, to help the authors make the manuscript easier to read and clearer for a broad scientific and engineering audience.

1. Overall, the supplement was a really useful document to actually understand how the experiments were performed, as there was not much information in the main document. One exception was for the particle tracing microrheology. The figure has a caption with information, but there was not a separate paragraph describing it. Something should be added to describe the equations used to find D from the MSD. Also, how many particles were used to find MSD? Does that matter? Or is it an average of the 7 or 10 experiments? Are the MSDs determined for each particle? Or are trajectories for the entire droplet concatenated and used as one trajectory?
2. Page 3, first paragraph (which started on page 2), something is logically not following for me. The authors label the LDH and see it partition. Then, they do measurements of kinetics and see more activity in the droplets compared to enzymes not in droplets and that proves that there is more enzyme in the droplets without labeling? But, since they haven't yet shown if the kinetics is affected by this crowding, I am not sure this experiment "proves" there is more enzyme in the droplets. I guess what I am thinking is that to show that there is more enzyme in the droplets, you would want a separate measure of protein concentration – not enzyme activity. Because at this point, you don't know yet if enzyme activity is the same or different in and out of droplets. I would suggest something lower-tech – like a SDS-Page gel. Since they can centrifuge and separate drops from the background, why didn't the authors just run these both on a gel. They could do it quantitatively with running a standard of known proteins and enzymes as a standard curve. Of course, they have a lot of protein, so they may need to dilute, but that is also still quantitative. They also use mass spec to figure out how much PEG is there for the phase separation. That might be another means to determine the enzyme concentration, no? Overall, the logic of using enzyme catalysis as a read out for enzyme concentration is at best not direct and at worse a logical fallacy.
3. Minor comment – recommend to start a new paragraph at the words, "With these results" in the first paragraph of page 3. This next section on the kinetics is just a different idea and needs to be its own paragraph. Not meaning to micromanage, but long, long paragraphs with too many ideas in them is hard for readers. This is especially true for such high level concepts as exist for this entire manuscript.
4. Page 3, paragraph 3, sentence, "By contrast, a simple LDH solution at the same metabolic rate

would consume all the pyruvate in less than 5 sec (Fig. 1f, inset).” I am not sure this sentence is correct. These two instances are at the same overall enzyme CONCENTRATION, but your results are showing that they have very different metabolic RATES. At least, that is what it looks like from Fig 1f and the figure caption, which says, “f, Sustained high metabolic rate for LDH (80 μ M inside the droplets) in presence of 5 mM NADH and 1 mM pyruvate. Inset: activity of 80 μ M LDH in solution in the presence of 2 mM NADH and 1 mM pyruvate.” Please correct this, as it is a very important point, and makes things confusing to use the wrong words here. Alternatively, if the word metabolic rate doesn’t mean enzyme reaction rate, then perhaps the phrase could be clarified?

5. In figure 2, the authors show that pH inside the droplet changes over time with the reaction in 2e and that the pH change depends on the droplet size in 2f. From 2f, the droplet pH could be anything from pH 7.2 to pH 9.0 depending on the size. Yet, the error bars in figure 2e, where the authors measure pH over time are very small. This makes me think that the time-dependence data was only measured for droplets of a certain size – most likely the largest ones where the pH is the highest – judging from the data in 2f. That should be made clear somewhere in the text or figure captions.

6. Throughout the manuscript, the authors show error bars, which is great. Unfortunately, they do not explain how many repeats they performed or if the error bars represent standard error of the mean or standard deviation. Sometimes, I could find more information in the supplement, but not always, and it was not always clear. Would it be possible to add that information in the relevant figure captions?

7. In figure 2f, the data for the individual droplets do not show uncertainty in the measurements. I assume these measurements have uncertainty, especially given the resolution of the microscope (x-axis data) and the calibration curve uncertainty from supplemental figure S6, so I am surprised that there is not uncertainty shown in that data. Perhaps it is very small, but again, that information needs to be given somewhere – maybe in the supplement at least? I am not saying the technique is bad or flawed – it is simply not perfect and has intrinsic measurement errors. Those need to be reported.

8. The modeling to try to understand the nature of the flow was interesting and fairly well-explained in the supplement. The supplement described three possible explanations for the driving of flows, but the main text only described two because one was estimated to be several orders of magnitude too small to explain the experimental results. From my reading, it looks like the two mechanisms, termed diffusiophoresis and marangoni would give velocities that are 16 μ m/s and 8 μ m/s, respectively. Both of these are at least 1 order of magnitude higher than the experimentally measured value. I don’t see much discussion of this discrepancy neither in the main text nor in the supplement. In the main text, the authors state: “In this approximation, the concentration of pH-determining species falls off roughly like $1/r$, and the predicted velocities from a pure surface-tension driven flow are 100-fold too large, as discussed in the Supplement. This, however, could be countered by diffusiophoretic flow to generate velocity scales comparable to those observed in the experiments, as shown by the simulation of the micro-pipette experiment in Fig. 3g.”

When going to the supplement, the discussion says, “Therefore, it appears that for the sizes of droplets in our experiments the bulk flow and the Marangoni flow are in direction competition.” These sentences are not really a discussion. From these sentences, I think the authors are saying that diffusiophoresis would set up flow in the opposite direction as Marangoni flow? Is that correct? Really, these velocities are vectors, so they would add vectorially, and there is no reason to assume the direction of the individual fields and there is not a clear directionality to me, as I read through the descriptions in either the main text or the supplement. I suppose it could be possible that the diffusiophoretic flow is in the opposite direction as the Marangoni flow, and maybe it was even stated somewhere, but I think I wasn’t given enough summary of these effects to understand that. I would appreciate a bit more explanation in the supplement, as promised.

9. Page 5, second paragraph, what does it mean to “ignore the action of the buffer”? Are you ignoring any flows in the buffer that might also affect the droplet?

10. Supplemental Figure S14, the authors used a terminal velocity method to determine the density difference between droplets. The figure in figure S14c is not a trajectory. It would be nice to see an overlay of the droplet images over time or something. What is being shown is just a single image of two droplets with two lines? I am not even sure what that represents? I am also unclear why the droplets are at different heights? Presumably, they were dropped the same distance, and the difference is due to them reaching terminal velocity at different times? Anyway, it is not the most important part of the paper, but it should be better explained.

Reviewer #2:

Remarks to the Author:

The work by Testa et al. demonstrated a phase-separated protein droplet system for the study enzyme activities that simulates the cytoplasm. With strongly partition of enzymes (LDH and urease) into protein-rich droplets, LDH assays show a significant increase in catalytic efficiency in the droplets, while pH gradients are generated by urease driving steady flow within droplets, mimicking cytoplasmic streaming. I find these results interesting and novel. I would therefore suggest revision at this stage as listed below.

1. Why can the host protein in solution form separate droplet phase in a polymer solution? What's the driving force? How about other proteins and polymers except BSA and PEG? Any criteria for the selection of proteins or polymers to construct such a system?
2. It is also mentioned that BSA and PEG components existed in both BSA-rich phase and PEG-rich phase. In page 1 paragraph 4, the author mentioned that "In our standard conditions, we prepare a solution with average concentrations of 232 mg/mL PEG and 37 mg/mL BSA", while the supporting information wrote "the final concentrations of the components were 230 mg/mL of PEG, 30 mg/mL BSA" in the droplet formation part. These descriptions are confused. Also, can the concentrations of each component be predetermined rather than measured?
3. Fig. 1a should be revised according to the real component distributions.
4. As the author described, the substrate can readily diffuse inside and product can diffuse out of the BSA-rich droplets. Why? What's the driving force?
5. The author shall check whether all dyes used for labelling was indicated.
6. For Fig. 1c, the starting time was not 0. Log[lactate concentration] rather than lactate concentration was used to make the plots. Why?
7. Could the author comment on the "substrate inhibition", because we generally mention "product inhibition" in reversible enzymatic reactions.
8. Why did LDH and urease have different kinetics in the droplets?
9. In terms of catalytic efficiency (k_{cat}/K_m), did the concentration range of Pyruvate considered? For example, the concentration of Pyruvate is 0-10 mM in Fig.1d and 0-1.5 mM in Fig.S3.
10. In Fig. 2c, the catalytic efficiency of urease in droplets was initially lower than that in buffer and higher after a certain time. Why?
11. Two mechanisms for the PH gradient-induced flow were introduced: a classical mechanism involving the surface, and a novel mechanism driven in the bulk. For the former one, how did the internal substances get influenced and form flow, though a surface tension factor was explored? The authors did not fully explain the second mechanism with experiment. Why?

Reviewer #1 (Remarks to the Author):

The manuscript Sustained Enzymatic Activity and Flow in Crowded Protein Droplets from Testa and colleagues is an exciting examination of the effects of crowding on enzymatic activity. Specifically, they show that when proteins are crowded with high levels of macromolecules so that they phase separate into liquid droplets, enzymes can partition into the droplets to have sustained activity. While the overall concentration of the enzyme in the test tube is low, the concentration within the droplets is higher. Further, the activity is sustained because of the portioning of the enzymes and the diffusion-limits for substrate to get to the proteins. Even more exciting, they show that these portioned enzymes can produce work and flows. These flows are driven by gradients created across the boundary of the droplets. Thus, the droplets confine proteins and small molecules without the need for a membrane. As the author's point out, these condensates could be the basis of early life and can also serve as platforms for chemical and materials innovation.

We thank Reviewer #1 for their insightful comments and for recognizing the significance of our findings on enzymatic activity within a crowded environment.

Positive aspects of the manuscript were:

- 1. Very exciting science and concepts*
- 2. Clear experimental results that were highly quantitative*
- 3. Modeling to accompany the experiments to help elucidate possible mechanisms of the flow induced in the droplets*

Some issues that I found with the manuscript were:

- 1. Some aspects of the manuscript were not clearly described or were described at too high a level for the broad audience of the journal*
- 2. Lack of information to assess reproducibility - repeats, error bars, fits, etc.*

Thanks for these constructive suggestions. We hope that we've addressed them to your satisfaction, as described below.

Below, I outline some of the issues I found, to help the authors make the manuscript easier to read and clearer for a broad scientific and engineering audience.

- 1. Overall, the supplement was a really useful document to actually understand how the experiments were performed, as there was not much information in the main document. One exception was for the particle tracing microrheology. The figure has a caption with information, but there was not a separate paragraph describing it.*

Are the MSDs determined for each particle? Or are trajectories for the entire droplet concatenated and used as one trajectory? Also, how many particles were used to find MSD? Does that matter? Or is it an average of the 7 or 10 experiments?

We very much appreciate Reviewer #1's careful reading of the supplement. To clarify our particle tracking methodology, we have added a panel to Fig. S1 and extended the supplemental text (paragraph *Particle tracking microrheology*).

Briefly, the MSDs are determined for each particle without concatenation. Occasionally, a single particle yielded more than one trajectory, if it went out of focus and reappeared at a later time within the acquisition window. This approach typically yielded about 400 tracks per droplet. In general, increasing the number of particles (and therefore the number of

tracks) to find the MSD increases the accuracy of the result. The roughly 400 tracks per droplet provided sufficient accuracy such that MSDs (obtained by averaging the roughly 400 tracks) between different droplets under the same conditions were similar, as evidenced in Fig. S1c.

Something should be added to described the equations used to find D from the MSD.

We used the following equation to derive the diffusion coefficient D from the MSD:

$$MSD = 4Dt,$$

and added it to the supplement as Eq. (7). We also clarified how the diffusion coefficient distribution and the ensemble diffusion coefficient for each time-lapse are calculated, and how the final value for each experimental condition is derived. We specified over which samples the diffusion coefficient was averaged to derive the final values of D and η . In this way we hope it is apparent to the reader which experiments are included in the calculations and how the curves in Fig. S1 are related to the data reported in the text.

2. Page 3, first paragraph (which started on page 2), something is logically not following for me. The authors label the LDH and see it partition. Then, they do measurements of kinetics and see more activity in the droplets compared to enzymes not in droplets and that proves that there is more enzyme in the droplets without labeling? But, since they haven't yet shown if the kinetics is affected by this crowding, I am not sure this experiment "proves" there is more enzyme in the droplets.

Note that we labelled LDH only for qualitative visualization, because we found that labeling strongly affects the enzyme activity (it decreases the activity more than 50 % compared to the unlabelled enzyme-data shown in the manuscript). Therefore, we performed every quantitative experiment using unlabeled enzymes.

The reviewer is correct. The original data shown in Fig 1c shows that enzyme activity is localized to droplet. It did not show that enzymes are localized to the droplets. We slightly adjusted the discussion to make this clearer.

A new experiment, described below, has been designed to quantify the relative enzyme concentrations.

I guess what I am thinking is that to show that there is more enzyme in the droplets, you would want a separate measure of protein concentration – not enzyme activity. Because at this point, you don't know yet if enzyme activity is the same of different in and out of droplets. I would suggest something lower-tech – like a SDS-Page gel. Since they can centrifuge and separate drops from the background, why didn't the authors just run these both on a gel. They could do it quantitatively with running a standard of known proteins and enzymes as a standard curve. Of course, they have a lot of protein, so they may need to dilute, but that is also still quantitative. They also use mass spec to figure out how much PEG is there for the phase separation. That might be another means to determine the enzyme concentration, no? Overall, the logic of using enzyme catalysis as a read out for enzyme concentration is at best not direct and at worse a logical fallacy.

This is a good suggestion. At the beginning of our study, we tried to visualize the enzyme using the SDS-PAGE. Unfortunately, LDH and BSA profiles on SDS PAGE are quite close to each other (around 40-45 and 55 KDa, respectively) and due to the high concentration of BSA in the droplets phase (as stated by the reviewer) even after dilution the quantification remains problematic. Furthermore, we found the presence of impurities from the vendor in

these proteins, that make the visualization but especially the quantification of the protein even more complicated.

To address the reviewers question, we designed and performed a new experiment to quantify partitioning.

In brief, we prepare the droplet and continuous phases equilibrated with a 3.3 μM LDH. Then, we separate and dilute the two phases 100 \times with buffer containing pyruvate and NADH. Note that with this dilution the droplets are fully dissolved. Then, we compare lactate production of the diluted droplet and supernatant phases, shown here:

Legend: Evaluation of LDH activity in buffer measuring lactate production after droplets dissolution (black) and after supernatant dilution (red) in buffer, in presence of 1 mM pyruvate and 2 mM NADH. The experiment has been performed in triplicate.

Looking at the first data point of this time course, it appears that the diluted droplet phase has about 200 \times the enzyme concentration than the equally diluted supernatant phase. However, just is the lower limit of the partitioning. Why? 1) we underestimate the rate of production from the diluted droplet phase, because all the substrate is consumed before the first data point 2) we overestimate the production in the diluted supernatant phase, because the initial recorded [lactate] values are at the resolution of the assay.

Since both phases are heavily diluted, the buffer conditions are very similar, and we expect the enzyme kinetic parameters to be the nearly the same. Therefore, the relative rate of lactate production is a good read-out of relative enzyme concentration of the original two phases.

Here is a detailed description of this new experiment and results.

- We have prepared 1 mL of LDH-containing droplets and we have centrifugated the sample for 30 minutes at 16900 g to divide the two phases.
- Then we separated the supernatant and the droplet phase in two different microcentrifuge tubes and we have then diluted both 1:2 with KP 0.1 M pH 7.0. In this way, we have diluted both samples, and **dissolved** the droplets.
- At this point we have prepared two different microcentrifuge tubes containing 0.98 mL of KP 0.1 M pH 7.0 with 2 mM NADH and 1 mM pyruvate and then started the reaction adding 20 μL of both samples to evaluate the enzymatic activity.

We have performed the same experiment for urease. Here is the data. In this case, we see an almost 1000x difference in the ammonia production rate.

Legend: Evaluation of urease activity in buffer measuring ammonia production after droplets dissolution (black) and after supernatant dilution (red) in buffer, in presence of 100 mM urea. The experiment has been performed in triplicate.

3. *Minor comment – recommend to start a new paragraph at the words, “With these results” in the first paragraph of page 3. This next section on the kinetics is just a different idea and needs to be its own paragraph. Not meaning to micromanage, but long, long paragraphs with too many ideas in them is hard for readers. This is especially true for such high level concepts as exist for this entire manuscript.*

We agree with Reviewer #1, and we started a new paragraph as suggested.

4. *Page 3, paragraph 3, sentence, “By contrast, a simple LDH solution at the same metabolic rate would consume all the pyruvate in less than 5 sec (Fig. 1f, inset).” I am not sure this sentence is correct. These two instances are at the same overall enzyme CONCENTRATION, but your results are showing that they have very different metabolic RATES. At least, that is what it looks like from Fig 1f and the figure caption, which says, “f, Sustained high metabolic rate for LDH (80 μ M inside the droplets) in presence of 5 mM NADH and 1 mM pyruvate. Inset: activity of 80 μ M LDH in solution in the presence of 2 mM NADH and 1 mM pyruvate.” Please correct this, as it is a very important point, and makes things confusing to use the wrong words here. Alternatively, if the word metabolic rate doesn’t mean enzyme reaction rate, then perhaps the phrase could be clarified?*

The main point of the plot in Figure 1f is to show that, with the same enzyme concentration, we can hugely increase the experimental window thanks to partitioning and dilution. Reviewer #1 has correctly pointed out an imprecision in the text, for which we are very thankful.

To clarify this, we have revised the main text accordingly. The sentence reported by Reviewer #1 now reads:

“By contrast, a simple LDH solution **at an enzyme concentration equal to that present in the droplets** would consume all the pyruvate in less than 5 sec (Fig. 1f, inset)”

Additionally, we have revised the Figure 1f caption to more directly reflect the plot and help explain the point in the main text. Now, instead of:

“Sustained high metabolic rate for LDH (80 μ M inside the droplets) in presence of 5 mM NADH and 1 mM pyruvate. Inset: activity of 80 μ M LDH in solution in the presence of 2 mM NADH and 1 mM pyruvate. [...]”

It reads:

“**Lactate production over time (equal to pyruvate consumption) for LDH in droplets (concentration 80 μ M inside the droplets) in presence of 5 mM NADH and 1 mM pyruvate. Inset: **Lactate production over time of LDH in buffer, at a concentration equal to that in droplets (80 μ M),** in the presence of 2 mM NADH and 1 mM pyruvate. [...]”**

Regarding the distinction between metabolic rate and reaction rate. They are linked through the enthalpy of reaction. In the supporting information, paragraph *Metabolic rate: calculation and measurement*, we write:

“The specific metabolic rate expresses the amount of energy per unit time, per unit volume generated by metabolic reactions. We linked this quantity to enzymatic parameters through the reaction rate

v and reaction enthalpy ΔH :

$$\dot{Q} = v \cdot \Delta H$$

[...] Considering that in most of our experimental conditions the substrate concentration is much larger than K_m , the reaction rate can be calculated from the estimated concentration of enzyme in the droplets $[E]_d$ and the enzyme turnover number k_{cat} as follows:

$$v = [E]_d \cdot k_{cat}$$

In principle, one would therefore expect to have the same metabolic rate whenever, given the same reaction, both the enzyme concentration and the turnover number are the same. In this specific case, while we can control the enzyme concentration and therefore ensure it is equal for the two experiments, we cannot assume that the turnover number is the same: firstly, because the enzyme is in two different environments, the droplets and the buffer, for which we have actually noted a small difference in the kinetic parameters. Furthermore, because the reaction in the buffer is so fast at 80 μ M enzyme concentration, the substrate is completely consumed faster than our sampling time, therefore any quantification of the kinetic parameters is impossible.

5. In figure 2, the authors show that pH inside the droplet changes over time with the reaction in 2e and that the pH change depends on the droplet size in 2f. From 2f, the droplet pH could be anything from pH 7.2 to pH 9.0 depending on the size. Yet, the error bars in figure 2e, where the authors measure pH over time are very small. This makes me think that the time-dependence data was only measured for droplets of a certain size – most likely the largest ones where the pH is the highest – judging from the data in 2f. That should be made clear somewhere in the text or figure captions.

We thank the reviewer for the comment. That is true, the pH inside the droplets could vary between pH 7.2 and 9.2 depending on the radius and urea concentration as shown in Fig. 2f. We agree with the reviewer’s comment, and we clarify it in the legend of Fig 2. as follows:

Now reads:

- e, Evaluation of the pH change over time for droplets with radius above 65 μm containing 1 μM urease at different urea concentrations. f, pH change inside the droplets (5 isolated droplets have been analyzed for each size interval) at 450 seconds at different urea concentrations and 1 μM urease. All error bars represent the standard deviation.

Instead of:

- e, f, Evaluation of the pH change over time and radius in the droplets containing 1 μM urease at different urea concentrations.

6. Throughout the manuscript, the authors show error bars, which is great. Unfortunately, they do not explain how many repeats they performed or if the error bars represent standard error of the mean or standard deviation. Sometimes, I could find more information in the supplement, but not always, and it was not always clear. Would it be possible to add that information in the relevant figure captions?

All the error bars in the plots represent the standard deviation. We added this information to the figures Fig. 1, Fig. 2, S3, S4, S5, S6, S7, S8. For the box plots in Fig. 3, S14 we specified what each of the uncertainty intervals indicates. We added the number of samples analyzed in the captions of Figure S16, and throughout the text in the paragraph "*Particle tracking microrheology*".

Everywhere else, either the curves are a result of only one experiment due to the difficulty of data acquisition, or the reported values are only an order-of-magnitude indication, and according to our judgement they are not quantitative enough to merit error bars.

7. In figure 2f, the data for the individual droplets do not show uncertainty in the measurements. I assume these measurements have uncertainty, especially given the resolution of the microscope (x-axis data) and the calibration curve uncertainty from supplemental figure S6, so I am surprised that there is not uncertainty shown in that data. Perhaps it is very small, but again, that information needs to be given somewhere – maybe in the supplement at least? I am not saying the technique is bad or flawed – it is simply not perfect and has intrinsic measurement errors. Those need to be reported.

Thank you. We have updated Fig 2f to include error bars.

8. The modeling to try to understand the nature of the flow was interesting and fairly well-explained in the supplement. The supplement described three possible explanations for the driving of flows, but the main text only described two because one was estimated to be several orders of magnitude too small to explain the experimental results. From my reading, it looks like the two mechanisms, termed diffusiophoresis and marangoni would give velocities that are 16 $\mu\text{m/s}$ and 8 $\mu\text{m/s}$, respectively. Both of these are at least 1 order of magnitude higher than the experimentally measured value. I don't see much discussion of this discrepancy neither in the main text nor in the supplement. In the main text, the authors state: "In this approximation, the concentration of pH-determining species falls off roughly like $1/r$, and the predicted velocities from a pure surface-tension driven flow are 100-fold too large, as discussed in the Supplement. This, however, could be countered by diffusiophoretic flow to generate velocity scales comparable to those observed in the experiments, as shown by the simulation of the micro-pipette experiment in Fig. 3g." When going to the supplement, the discussion says, "Therefore, it appears that for the sizes of droplets in our experiments the bulk flow and the Marangoni flow are in direction competition." These sentences are not really a discussion. From these sentences, I think the authors are saying that diffusiophoresis would set up flow in the opposite direction as Marangoni flow? Is that correct?

Really, these velocities are vectors, so they would add vectorially, and there is no reason to assume the direction of the individual fields and there is not a clear directionality to me, as I read through the descriptions in either the main text or the supplement. I suppose it could be possible that the diffusiophoretic flow is in the opposite direction as the Marangoni flow, and maybe it was even stated somewhere, but I think I wasn't given enough summary of these effects to understand that. I would appreciate a bit more explanation in the supplement, as promised.

We agree with the reviewer that further explanation for the differences between the Marangoni effect and the bulk phoresis effect is needed. For the Marangoni effect, the velocity across the droplet interface must remain continuous and that gives rise to a large discontinuity in the interfacial stresses due to the effect of surface tension. On the other hand, for bulk phoresis, the interfacial stresses remain unchanged at the droplet boundary which requires a velocity jump at the interface. Now for an axisymmetric droplet, such as the one considered in this model, these two processes result in opposite flows inside the droplet: the Marangoni effect creates a flow that drives the fluid towards the pipette from center of the droplet, while for the bulk phoresis-driven flow, the fluid flows away from the pipette at the center line. That is why the competition of these two effects can generate a flow inside the droplet that scales similarly to our experimental observations. We have added this explanation to the SI (page 26, second to last paragraph).

We also now mention the third mechanism (surface phoresis) in the revised manuscript, and discuss why it can be safely neglected in our analysis. (page 4, last paragraph).

9. Page 5, second paragraph, what does it mean to "ignore the action of the buffer"? Are you ignoring any flows in the buffer that might also affect the droplet?

With this sentence we mean that we neglect the capacity of the buffer of neutralizing the alkaline/acidic ions and to maintain the pH at a fixed value. We revised the sentence to make it more explicit. It now reads:

"For simplicity, we focused on the micro-pipette experiment and ignored the action of the buffer in stabilizing the pH value."

10. Supplemental Figure S14, the authors used a terminal velocity method to determine the density difference between droplets. The figure in figure S14c is not a trajectory. It would be nice to see an overlay of the droplet images over time or something. What is being shown is just a single image of two droplets with two lines? I am not even sure what that represents? I am also unclear why the droplets are at different heights? Presumably, they were dropped the same distance, and the difference is due to them reaching terminal velocity at different times? Anyway, it is not the most important part of the paper, but it should be better explained.

This is an excellent suggestion that significantly improves the Supplementary Figure. We revised panel c of Figure S14 adding a time-series image of falling droplets overlapped on the tracked trajectories. We also revised the Figure caption to better reflect what it currently shows, and we added a sentence to clarify that the lines in panel c represent the tracked trajectories. It now reads:

"Images of two falling droplets, overlaid to the respective trajectories resulting from tracking (blue and yellow lines). The time-series image consists of 10 overlapped frames, 8.4 seconds apart from each other (the left droplet exits the frame after 68 seconds, at time frame 9)."

The height difference between the droplets depends on two factors:

1. The droplets have different limit velocities (61.2 $\mu\text{m/s}$ and 58.6 $\mu\text{m/s}$ respectively, from Fig. S14d). Since the dropping point of the droplets is roughly 2.5 centimeters above

than the observed frame, by the time they enter the frame they accumulate a significant vertical separation.

2. They are dropped at slightly different times.

While the dropping point being so far away from the observed frame has the inconvenience of introducing this height difference, it has the significant advantage of ensuring most of the droplets observed have reached their limit velocity, which is an essential requirement for the reliability of this measurement.

We have added a sentence to the caption to justify the height difference:

“The height difference between the droplets in the frame arises from two factors: the different terminal velocity reached, which results in vertical separation accumulated from the dropping point (out of frame), and the time delay present between the release of different droplets.”

Reviewer #2 (Remarks to the Author):

The work by Testa et al. demonstrated a phase-separated protein droplet system for the study of enzyme activities that simulates the cytoplasm. With strong partitioning of enzymes (LDH and urease) into protein-rich droplets, LDH assays show a significant increase in catalytic efficiency in the droplets, while pH gradients are generated by urease driving steady flow within droplets, mimicking cytoplasmic streaming. I find these results interesting and novel. I would therefore suggest revision at this stage as listed below.

Thank you.

1. Why can the host protein in solution form a separate droplet phase in a polymer solution? What's the driving force?

The driving force for the phase separation of BSA by the action of PEG is the so-called *depletion force*, which was first described by Asakura and Oosawa in 1954¹. The general description is based on osmotic pressure and excluded volume.

How about other proteins and polymers except BSA and PEG? Any criteria for the selection of proteins or polymers to construct such a system?

The power of depletion interactions is that they only depend on the relative size and concentration of the two species in solution. This allows the choice of a wide variety of molecules for both depletant and depleted species. PEG, alginate, dextrans and more have been reported as effective depleting agents, while a plethora of proteins, DNA, RNA have been successfully phase separated from solution using them^{4,5}. However, not all the combinations yield to phase separated phases that are liquid-like, so one would have to play with relative component concentrations, salts and buffer concentrations to try to achieve that. We performed some preliminary experiments involving proteins other than BSA, some more successful than others in yielding to a liquid-like deposit.

Regarding our particular system, PEG was selected here because it is the most used depletion agent² owing to its low price, and the fact that it is available in a wide range of molecular weights (from a few hundreds to millions of Daltons). This allows for great flexibility in its size choice, which is extremely important for depletion interactions. Additionally, it is the most used crowding agent and its ability to greatly concentrate macromolecules and

1 Sho, A., & Fumio, O. (1954). On Interaction between Two Bodies Immersed in a Solution of Macromolecules. *J. Chem. Phys.* 22, 1255 (1954), 22(7), 1255–1256.

2 Sapir, L., & Harries, D. (2015). Is the depletion force entropic? Molecular crowding beyond steric interactions. *Current Opinion in Colloid and Interface Science*, 20(1), 3–10. <https://doi.org/10.1016/j.cocis.2014.12.003>

proteins have been also widely reported³. Since crowding is an essential characteristics of bacteria cytoplasm, this was an important feature we wanted to replicate. We chose BSA as a model protein for its robustness, relatively low price, and a molecular weight that is very close to the average of the macromolecules in the cytoplasm of *E. coli*.

2. *It is also mentioned that BSA and PEG components existed in both BSA-rich phase and PEG-rich phase. In page 1 paragraph 4, the author mentioned that “In our standard conditions, we prepare a solution with average concentrations of 232 mg/mL PEG and 37 mg/mL BSA”, while the supporting information wrote “the final concentrations of the components were 230 mg/mL of PEG, 30 mg/mL BSA” in the droplet formation part. These descriptions are confused. Also, can the concentrations of each component be predetermined rather than measured?*

While in the main text we report the average concentrations measured over a series of experiments, in the supporting information we report the target concentrations we used in our “recipe” for droplets making. We clarified this distinction rephrasing the supporting information, which now read:

“At the standard working conditions **we mixed the components with a final target concentration of 230 mg/mL of PEG, 30 mg/mL BSA [...]**”

3. *Fig. 1a should be revised according to the real component distributions.*

We have applied the following changes to the Fig. 1a:

- We changed how we represent polymer molecules to a simple circle to enhance readability.
- We have increased the number of polymer chains in the polymer-rich phase to more closely reflect the concentration ratios reflect the fact that its concentration is, in most cases, quite high
- We rephrased the figure caption as follows:

“Schematic representation of **a general** active liquid-liquid phase separated protein droplets with partitioned enzyme, **by action of a crowding polymer.**”

4. *As the author described, the substrate can readily diffuse inside and product can diffuse out of the BSA-rich droplets. Why? What’s the driving force?*

Small molecules do not feel the depletion force. Therefore, they can diffuse freely according to Fick’s Law.

More detailed discussions of the diffusion of substrate and product are found in the “*Transport limitations*” section of the supporting information.

5. *The author shall check whether all dyes used for labelling was indicated.*

We thank the Reviewer #2 for the comment, which made us notice we forgot to report the labelling dye of BSA in Figure 1a. We added that information to the supplement, paragraph “*Protein labelling*”.

3 Akabayov, B., Akabayov, S. R., Lee, S. J., Wagner, G., & Richardson, C. C. (2013). Impact of macromolecular crowding on DNA replication. *Nature Communications*, 4. <https://doi.org/10.1038/ncomms2620>

6. For Fig. 1c, the starting time was not 0. Log[lactate concentration] rather than lactate concentration was used to make the plots. Why?

The time axis of 1c starts at $t=0$.

The vertical axes in 1c and 2b are chosen to be logarithmic because the rates of lactate production are so different in the two cases. At $t=0$, lactate production is about 100x higher. If we adjusted the linear axis to clearly show production with droplets, you would not resolve the supernatant production.

7. Could the author comment on the “substrate inhibition”, because we generally mention “product inhibition” in reversible enzymatic reactions.

The pyruvate (substrate) inhibition of LDH is a well-known mechanism studied and reported in the literature since 1965⁴ ⁵. On the other hand, the same is valid for the product inhibition⁶. In our work, we are considering substrate inhibition because we are evaluating the activity of LDH varying the concentration of the substrate. We did not use lactate (product) in our kinetics study to test LDH activity or inhibition, therefore we cannot comment about product inhibition.

8. Why did LDH and urease have different kinetics in the droplets?

Confinement to droplets has a significant effect on the k_{cat} of LDH and K_m of urease. We have not isolated the underlying mechanisms for these changes. Very generally, you might expect k_{cat} to be lowered by the high viscosity inside a droplet, and K_m to be altered by crowding or competitive binding. We are currently performing systematic studies to get to the bottom of this important question.

9. In terms of catalytic efficiency (k_{cat}/K_m), did the concentration range of Pyruvate considered? For example, the concentration of Pyruvate is 0-10 mM in Fig.1d and 0-1.5 mM in Fig.S3.

This is a nice observation. In terms of catalytic efficiency (k_{cat}/K_m , k_{cat} and K_m values) we considered and presented in the manuscript the values extrapolated from the plot of Fig S3 considering pyruvate concentration from 0 to 1.3 mM (as explained in caption Fig.1). We decided to calculate the Michael Menten parameters using the classic Michael Menten equation without substrate inhibition. However, we also wanted to show to the readers the substrate inhibition profile to demonstrate that qualitatively LDH behaves kinetically as in buffer. We clarify this in the manuscript as follows:

AS IT WAS:

- Michaelis-Menten parameters are reported in the inset of Fig. 1d

NOW IT READS

⁴ Vesell and Elliot S. "Lactate dehydrogenase isozymes: substrate inhibition in various human tissues." *Science* 150.3703 (1965): 1590-1593.

⁵ Kaplan, et al "Significance of substrate inhibition of dehydrogenases." *Annals of the New York Academy of Sciences* 151.1 (1968): 400-412.

⁶ Karlsson, Jan, Bodil Hulten, and Bertil Sjodin. "Substrate activation and product inhibition of LDH activity in human skeletal muscle." *Acta Physiologica Scandinavica* 92.1 (1974): 21-26.

- *Michaelis-Menten parameters are reported in the inset of Fig. 1d (obtained from the fitting reported in Fig. S3, pyruvate concentration between 0 and 1.3 mM)*

10. In Fig. 2c, the catalytic efficiency of urease in droplets was initially lower than that in buffer and higher after a certain time. Why?

Please note that the values (V , sec^{-1}) in Fig. 2c are plotted against substrate concentration not over time. The reaction rate increases more quickly in buffer at low urea concentrations, because urease has a lower K_m in buffer. At large concentrations, the reaction rates are faster in droplets because the urease has a higher k_{cat} in the droplets. (see Michaelis-Menten parameters in inset of 2c)

11. Two mechanisms for the PH gradient-induced flow were introduced: a classical mechanism involving the surface, and a novel mechanism driven in the bulk. For the former one, how did the internal substances get influenced and form flow, though a surface tension factor was explored? The authors did not fully explain the second mechanism with experiment. Why?

The two mechanisms have the interesting feature that they arise from different geometric features of the droplet. The surface tension gradient introduces a surface drive and the phoretic drive introduces a bulk flow. One thing that they both have in common, however, is that they arise as a direct consequence of the composition of our dense liquid droplets; namely, the bulk effect arises due to the high density of the enzymes in the bulk of the droplets and the surface-tension arises due to the appearance of an interface as result of the depletion-force-induced condensation. Consequently, it will be difficult to recreate the conditions for the new bulk effect in a controlled experiment without creating an interface (that will bring in the Marangoni effect) at the same time. This is why in this investigation we were largely reliant on experimental probes of the surface tension effect combined with theoretical comparison of the two mechanisms in combination with our measurements of the flow properties using values for the surface tension from our experiments.

Reviewers' Comments:

Reviewer #1:

Remarks to the Author:

The authors have updated the manuscript to my satisfaction.

Reviewer #2:

Remarks to the Author:

The authors have responded to the reviewers' comments and conducted additional experiments to support their arguments. I would accept publication.